# Interplay Between HIV and Human Pegivirus (HPgV) Load in Co-Infected Patients: Insights from Prevalence and Genotype Analysis

**DOI:** 10.3390/v16010005

**Published:** 2023-12-19

**Authors:** Muammer Osman Köksal, Martin Pirkl, Kutay Sarsar, Mehmet Ilktaç, Gibran Horemheb-Rubio, Murat Yaman, Sevim Meşe, Haluk Eraksoy, Baki Akgül, Ali Ağaçfidan

**Affiliations:** 1Department of Medical Microbiology, Istanbul Faculty of Medicine, Istanbul University, Istanbul 34093, Türkiye; kutay.sarsar@istanbul.edu.tr (K.S.); murat.yaman@istanbul.edu.tr (M.Y.); smese@istanbul.edu.tr (S.M.); ali.agacfidan@istanbul.edu.tr (A.A.); 2Institute of Virology, Faculty of Medicine and University Hospital of Cologne, University of Cologne, 50935 Cologne, Germany; martin.pirkl@uk-koeln.de (M.P.); gibran.rubio@pei.de (G.H.-R.); baki.akguel@uk-koeln.de (B.A.); 3Department of Pharmaceutical Microbiology, Faculty of Pharmacy, Eastern Mediterranean University, Famagusta 99450, Cyprus; mehmet.ilktac@emu.edu.tr; 4Department of Infectious Disease and Microbiology, Istanbul Faculty of Medicine, Istanbul University, Istanbul 34093, Türkiye; eraksoyh@istanbul.edu.tr

**Keywords:** human immunodeficiency virus (HIV), human pegivirus (HPgV) prevalence, HIV/HPgV co-infection

## Abstract

Human pegivirus (HPgV) is transmitted through sexual or parenteral exposure and is common among patients receiving blood products. HPgV is associated with lower levels of human immunodeficiency virus (HIV) RNA and better survival among HIV-infected patients. This study aimed to investigate the prevalence of HPgV and determine its subtypes in HIV-infected individuals living in Istanbul, which has the highest rate of HIV infection in Türkiye. Total RNA extraction from plasma, cDNA synthesis, and nested PCR were performed for HPgV on plasma samples taken from 351 HIV-1-infected patients. The HPgV viral load was quantified on HPgV-positive samples. HPgV genotyping was performed by sequencing the corresponding amplicons. In the present study, the overall prevalence of HPgV RNA in HIV-infected patients was 27.3%. HPgV subtypes 1, 2a, and 2b were found, with subtype 2a being the most frequent (91.6%). Statistical analysis of HIV-1 viral load on HPgV viral load showed an opposing correlation between HIV-1 and HPgV loads. In conclusion, these data show that HPgV infection is common among HIV-positive individuals in Istanbul, Türkiye. Further comprehensive studies are needed to clarify both the cellular and molecular pathways of these two infections and to provide more information on the effect of HPgV on the course of the disease in HIV-infected individuals.

## 1. Introduction

Human pegivirus (HPgV, formerly GBV-C/HGV) is a single-stranded, positive-sense RNA virus and characterized by a genomic size of approximately 9.4 kb. HPgVs belong to the genus Pegivirus in the family Flaviviridae and are among the most prevalent flaviviruses worldwide [1]. Upon examination of its genome organization and sequence homologies, HPgV shows its closest genetic relationship to human hepatitis C virus (HCV), which also belongs to the Flaviviridae family. However, unlike HCV, HPgV does not exhibit hepatotropic tendencies nor the concomitant subsistence of antibodies and viremia. The clinical significance of HPgV remains incompletely understood [2]. Infection with HPgV is usually asymptomatic and is generally considered non-pathogenic. However, controversial research has suggested a possible association between HPgV infection and diseases such as lymphoma and various extrahepatic diseases. It is worth noting that these hypotheses require further substantiating evidence for confirmation [3]. The prevalence of HPgV viremia is 1–5% among healthy blood donors in the U.S. and Western Europe, but higher frequencies have been observed in developing countries (up to 21.7%) [4,5,6]. Epidemiological data has shown that HPgV is common among populations worldwide. HPgV is further classified into seven genotypes and subtypes. The distribution of the HPgV genotypes varies across countries: genotype 1 mainly in West Africa, genotype 2 in North America and Europe, genotype 3 in Asia and Latin America, genotype 4 in South East Asia, especially Vietnam, genotype 5 in South Africa, and genotype 6 predominantly in Indonesia. Genotype 7 has been described in the Chinese province of Yunnan [7,8]. Such diversity of genotypes is associated with the genetical background of the infected subjects and highlights the complexity of HPgV epidemiology. HPgV is considered a non-pathogenic human virus, transmitted primarily through sexual contact, parenteral exposure (e.g., blood transfusions), and vertical transmission from mother to child. It is particularly common among individuals infected with the human immunodeficiency virus (HIV) [5,9]. It has been reported to be associated with decreased morbidity and mortality in HIV-infected patients [5,10,11]. HIV-1/HPgV-co-infected patients showed a better response after antiretroviral therapy (ART), and a decrease in HIV-1 viral load was observed [5,12]. The worldwide prevalence of HPgV in HIV-positive individuals ranges from 15% to 41% for the HIV patients [5]. These rates varied according to the patient group and region where the study was conducted [13,14,15,16]. Early studies suggested that HPgV-co-infected HIV-positive individuals provide a favorable clinical outcome, delaying the progression to AIDS, compared to patients infected with HIV-1 alone [5,10,11,17]. It was also observed that in these individuals there was a decrease in HIV-1 viremia and an increase in the number of CD4 + T lymphocytes (CD4 + TL). These improvements were observed independently of antiretroviral use, age, ethnicity, gender, or HIV-1 transmission route [5,11,12]. Studies have shown that HPgV could act on HIV infection by either directly interfering with viral entry or indirectly regulating host factors that improve replication and disease progression [18]. Either way, the mechanisms of this interaction still unravel several factors that may cause delays in HPgV-induced disease progression. HPgV has been suggested to regulate the cytokine response, as HIV infection, which generally stimulates the Th2 response, stimulates the Th1 response in the presence of HPgV, enhancing the antiviral response to HIV infection [19]. In vitro, HPgV infection increases the production of certain chemokines, such as RANTES, macrophage inflammatory protein (MIP) -1α, MIP-1β, and SDF-1, while decreasing CCR5 expression on the cell surface [20]. The number of people infected with HIV has risen significantly in recent years, and HIV is a public health problem in Türkiye [21]. This study aimed to determine the frequency of HPgV/HIV-1 co-infection in Istanbul, and the genotypes of HPgV present in these patients, in a cross-sectional and longitudinal study.

## 2. Materials and Methods

### 2.1. Study Population

A total of 351 HIV-infected adult patients who applied for routine plasma HIV-1 RNA viral load test and CD4/CD8 T lymphocyte counts between January 2020 and June 2020 in the Department of Medical Microbiology, Istanbul Faculty of Medicine, were included in this study. Patients were divided into two categories: treatment-naïve patients (*n* = 71), representing those who had never received HIV-1 treatment, and treatment-experienced patients (*n* = 280), representing patients who had received at least one treatment. Thirteen treatment-naive patients who showed a high HIV-load (*n* = 13) were further characterized in a longitudinal study for HPgV load by taking two further samples one and three months after the first blood sample. The longitudinal samples were tested for HIV-load, HPgV load, and CD4/CD8 counts.

### 2.2. Collection of Samples

Ten milliliters of blood were collected by venipuncture in tubes containing EDTA. Aliquots of the plasma and leukocyte fractions were separated by centrifugation at 2500 rpm for 15 min and stored in 2.5 mL tubes at −70 °C until use.

### 2.3. Quantification of HIV-1 Plasma Viral Load and C4/CD8 Cell Count

HIV-1 viral load was initially measured using the Artus HI Virus-1 QS-RGQ kit on the Qiasymphony RGQ system (Qiagen GmbH, Hilden, Germany), with a detection limit of 45 copies/mL (measurement range between 45 and 45,000,000 copies/mL). Lymphocyte measurements were performed using flow cytometer (FACScountTM reagents, Becton-Dickinson, San Jose, CA, USA). Both the methods followed the manufacturer’s recommendations.

### 2.4. HPgV Detection

Total nucleic acids isolated from the plasma for qualitative HIV-1 RNA tests were used in this study. The studies were performed on the same day, considering the possible decrease in RNA levels. The amount of RNA in the isolated total nucleic acid was measured using a NanoVue™ Plus Spectrophotometer (GE Healthcare, Buckinghamshire, UK), and samples were diluted with sterile distilled water to 50 ng RNA per reaction. HPgV RNA in plasma was first transformed into DNA and amplified by reverse transcription-polymerase chain reaction (RT-PCR) using the One-Step RT-PCR Kit (Qiagen GmbH, Hilden, Germany). RT-PCR reaction from 2 μL extracted RNA in an amplification reaction mix containing 5 μL 10 × OneStep RT-PCR Buffer, 2 μL 10 mM dNTP mix, 2 μL OneStep RT-PCR Enzyme Mix, 1 μL of a 0.6 µM solution of each primer (HG1 and HG1R) was formed. The PCR mix was adjusted to a final volume of 50 µL by adding RNAse/DNAse-free water. The nested PCR, comprising a two-step PCR, was directed to the 5′UTR region of HPgV. The first PCR external primer pair was HG1 (5 -GGTCGTAAATCCCGGTCACC-3′, sense) and HG1R (5′-CCCACTGGTCCTTGTCAACT-3′, antisense), corresponding to the nucleotide 139 nt to 158 nt and 381 nt to 400 nt on the reference accession number AF121950. The PCR reaction conditions were maintained for 30 min at 50 °C, followed by 15 min of hot-start Taq activation at 95 °C, followed by 40 cycles of 1 min at 94 °C, 1 min at 56 °C, and 1 min at 72 °C. The reaction was terminated by incubation at 72 °C for 15 min for final extension. For nested PCR, 25 μL of reaction mix, 0.75 units of Taq DNA polymerase, 5 mM dNTPs, 15 mM MgCl_2_, 25 pmol of each primer, and 10 μL of MyTaq ™ DNA polymerase (Bioline Ltd., London, UK) were used. Genomic DNA (2 μL, 150–250 ng) was added, followed by initial denaturation at 95 °C for 5 min, and 35 cycles of 95 °C for 45 s, 58 °C for 45 s, and 72 °C for 45 s. The final extension step was performed at 72 °C for 10 min. The second amplification was performed using the internal primer pair HG2 (5-TAGCCACTATAGGTGGGTCT-3′, sense) and HG2R (5′-ATTGAAGGGCGACGTGGACC-3′, antisense), corresponding to the nucleotide 163 nt to 182 nt and 331 nt to 350 nt on the reference accession number AF121950 [22]. As previously described, this 5′ UTR region contains not only several blocks of well-conserved sequences that can be useful in screening for HPgV PCR assay but also genotype-specific sequences [22]. The PCR products were analyzed by electrophoresis on a 4% agarose gel. In each PCR test, one negative control and one positive control were tested in addition to the samples of interest.

### 2.5. HPgV Sequencing and Phylogenetic Analysis

The nested PCR products were purified using the QIAquick^®^ PCR Purification Kit (Qiagen GmbH, Hilden, Germany). Sequencing of the purified PCR products was performed using the aforementioned primer pairs and the Big Dye Terminator Cycle Sequencing Ready Reaction Kit (Applied Biosystems, Foster City, CA, USA). The PCR products were analyzed using an ABI PRISM 3730 automatic sequencing device (Applied Biosystems, Foster City, CA, USA). The obtained sequences were analyzed using BioEdit and MEGA 6.0. BLAST and ClustalW were used to select the sequences from GenBank. Phylogenetic trees were generated using TreeView and DNA Maximum Likelihood program in Mega 6 and aligned with the reference sequences of the seven main genotypes obtained from GenBank (AB013500, U36380 (Genotype 1); D90600, AY196904, U44402, U45966, LT009485, LT009494, AF104403 (Genotype 2; 2a); U63715 (Genotype 2; 2b); D87713, D90601 (Genotype 3); AB0188667 (Genotype 4); AY949771 (Genotype 5); AB003292 (Genotype 6); and HQ331235 (Genotype 7)).

### 2.6. HPgV Quantification

HPgV viral load was quantified using One-Step RT-Real-Time PCR (Qiagen GmbH, Hilden, Germany) with a detection limit of 50 copies per milliliter. The process involved the following steps: Reverse transcription at 50 °C for 30 min, DNA polymerase activation at 95 °C for 2 min and PCR amplification including 40 cycles. Each cycle comprised denaturation at 95 °C and annealing at 55 °C. Specific primers targeting the 5′ untranslated region (103–163 nt) from the GenBank accession number AF121950 were used. These included the sense primer (5′-GGC CAA AAG GTG GTG GAT GG-3′, 103–122 nt) and the antisense primer (5′-CTT AAG ACC CAC CTA TAG TGG CTA CC-3′, 188–163 nt). DNA polymerase extended the primers, and real-time detection was enabled using a TaqMan probe. This probe contained a fluorophore (FAM) and a quencher (TAMRA) (131–158 nt), 5′-(FAM) TGA CCG GGA TTT ACG ACC TAC CAA CCC T (TAMRA)-3′. A puc57 plasmid containing the PCR-amplified 5′UTR HPgV region (85–362 nt) was used as a reference standard curve, spanning concentrations from 10^2^ to 10^9^ copies [23].

### 2.7. Ethical Statements

All subjects gave their informed consent for inclusion before they participated in the study. The study was conducted in accordance with the Declaration of Helsinki, and the protocol was approved by the Research Ethics Committee of the Istanbul University, Istanbul Faculty of Medicine (Approval number: 917/2020).

### 2.8. Statistical Analysis

The skewed viral load distributions of HIV-1 and HPgV were normalized by logarithm with base 10, and linear regression on HIV-1 viral load (R package ‘stats’ (version 4.2.2) was performed [24]. The co-variables of age, CD4 count, CD8 count, and CD4/CD8 ratio were included in the linear model of the cross-sectional samples. For the analysis of the longitudinal samples, the three time points for each patient were combined and added as a factor in the regression analysis. Partial correlation was calculated using the inverse precision matrix. Independence was tested using the Fisher transformation. HIV-1 and HPgV loads were binarized using hierarchical clustering. The percentage and mean values with standard deviations were calculated. Significance was tested using Fisher’s exact test and a chi-squared test. The level of significance was set at α = 0.05, and multiple testing calculations (Table 1) were corrected with Bonferroni.

## 3. Results

In the cross-sectional study, HPgV was detected in 96 patients (27.3%) of a total number of 351 HIV-infected patients. All HPgV-positive patient samples were subsequently sequenced. Phylogenetic analysis revealed that genotypes 1, 2a, and 2b had frequencies of 1.05% (1/96), 92.7% (89/96), and 6.25% (6/96), respectively (Figure 1). The demographic and laboratory data of HIV-infected and HIV/HPgV-co-infected patients are shown in Table 1.

When the age data of the two groups were examined, no significant differences were observed. HIV-1-infected patients who participated in this study encompassed a broad age range spanning 17 to 76 years, with a calculated mean age of 43.5 years. Conversely, HIV/HPgV-co-infected patients were between 22 and 70 years of age with a slightly younger mean age of 41.9 years. Furthermore, when considering the gender distribution within the cohort, a notable contrast emerged in the context of HIV/HPgV co-infection. Males exhibited a significantly higher prevalence of co-infection compared to their female counterparts, with a striking disparity of 29.2% among males versus 12.8% among females (*p* < 0.05). Another important aspect of this study concerned the viral load in these patient groups. Of note, HPgV-positive patients (*n* = 96) had a lower viral load, with a calculated mean HIV-1 viral load of 4.89 log copies of RNA per milliliter, compared to their HPgV-negative counterparts (*n* = 255), who had a mean viral load of 5.28 log copies of HIV-1 RNA per milliliter (*p* < 0.05). The rates of HPgV-1 infection among individuals infected with HIV-1 did not exhibit a significant age-related difference in the age groups 16–30 years (28.6%), 31–50 years (27.1%), and ≥ 51 (27.1%) (*p* > 0.05). Additionally, when examining the CD4+ T lymphocyte counts, the HPgV-positive group exhibited higher counts compared to the HPgV-negative group, with values of 486 vs. 399, and this difference was statistically significant (*p* < 0.05) with a strong disproportion between cell counts <200 (9.8%) and ≥200 (32.7%). In relation to disease progression, the prevalence of AIDS was significantly higher in patients with HIV-1 mono-infection. It is important to note that there were no statistically significant differences in CD8+ T lymphocyte counts between the groups. Interestingly, HIV viral load was significantly higher in mono-infected patients not receiving antiretroviral therapy (ART). ART use was not significant between HPgV-positive and -negative patients with 29.6% of HPgV-positive patients receiving and 18.3% of HPgV-positive not receiving ART. CD4+ T lymphocyte counts were significantly higher in both mono-infected and co-infected patients receiving ART. Irrespective of the use of ART, the viral load in the HIV-1/HPgV-co-infected group was significantly lower than in the mono-infected group. It is worth mentioning that due to the limited number of patients meeting the AIDS definition in the HIV-1/HPgV-infected patient group, a statistical comparison could not be made in this regard. However, when conducting a linear regression analysis of HIV-1 viral load while considering HPgV viral load, age, CD4 count, CD8 count, and CD4/CD8 ratio, a significant (*p* < 0.001) correlation between HIV-1 and HPgV viral load was identified (Figure 2, left). Some HIV-1 and HPgV load pairs showed a dichotomous, almost binary pattern. We account for this by hierarchical clustering of both viral loads with a cutoff for two classes and performed Fisher’s exact test and a chi-squared test (*p* < 0.001). This confirmed the inverse relationship of both viral load distributions.

The time series showed a similar but not significant (*p* > 0.1) trend, but with an even more dichotomous viral load distribution relative to the number of data points (Figure 2, right). This is not unexpected because the time point itself already accounts for decreasing HIV-1 and increasing HPgV loads and can remove causal independence. This means that a third confounder might be causally responsible for both viral loads. However, the partial correlation between HIV-1 and HPgV viral loads was significant enough (*p* < 0.05) to reject the hypothesis of independence. Hence, given the time-point, age, CD4 count, CD8 count, and CD4/CD8 ratio, HIV-1 and HPgV viral load showed a significant dependency. This also confirms our explanation with a third confounder, using the variable time point as a proxy. Thus, it is not known why HIV-1 viral load decreases and why HPgV load increases over time. Hierarchical clustering was used to fully binarize the already dichotomous distributions of HIV-1 and HPgV loads. Both Fisher’s exact and chi-squared tests showed that the distribution in high and low viral loads of both viruses was significantly (*p* > 0.01) negatively correlated over time. However, this negative relationship was not observed in one patient (sample ID 94). It is believed that the observed difference between the HPgV viral load and HIV-1 RNA viral load in that particular patient, unlike in the other patients, is most probably attributable to either the non-use or suboptimal use of antiretroviral medication (Figure 3 and Appendix A). This is further confirmed by the decreasing CD4 cell count (green line).

## 4. Discussion

Within the scope of this research, we assessed the prevalence, genotypic distribution, and potential clinical implications of HPgV infection on HIV disease progression among a cohort of 351 HIV-infected patients in Istanbul, Türkiye. The results of this study revealed an overall positive detection rate of HPgV-1 in HIV-infected patients at 27.3%. This rate falls within the range reported in similar studies conducted worldwide, where the prevalence of HPgV/HIV-1 co-infection has been documented to vary between 11% and 40% among HIV patients [11,25,26]. These differences in frequency may be attributed to the distinct dynamics of transmission within each population. In addition, the presence of other epidemiological components and differences in the sensitivity and specificity of the diagnostic methods used may play a role in the differences in frequencies. While some prior studies have suggested a significantly higher prevalence of HPgV infection among younger patients [27], our study did not observe a statistically significant difference in prevalence between different age groups. It is worth noting that cross-sectional studies conducted in Türkiye have reported a prevalence range of 4.1% to 14% in various subpopulations, including blood donors, hemodialysis patients, and other high-risk groups [28,29,30]. Therefore, the current finding of an HPgV prevalence of 27.3% among HIV-1-infected patients is relatively high compared to other studies involving diverse patient groups in Türkiye. This difference in prevalence may be attributed to the fact that HIV-infected patients may exhibit risky behaviors more frequently. Additionally, both HIV-1 and HPgV share common transmission routes, primarily utilizing sexual and parenteral routes for host entry [2].

Studies on the genetic diversity of HPgV revealed the presence of seven different genotypes at the global level, with significant regional differences [18,31]. The results of the present phylogenetic analysis revealed the predominant presence of genotype 2, including subtypes 2a and 2b, in the population studied. In contrast, genotype 1 was found in only one case. These results are consistent with those observed in different regions of Europe, where the same genotypes have been documented in the HIV-1-infected population [4,31]. Numerous epidemiologic studies have highlighted the potentially beneficial effects of HPgV viremia on disease progression in HIV-1-infected individuals. CD4+ T cell count and HIV-1 viral load are two important predictors of HIV/AIDS disease progression and are used to determine the initiation and evaluate the efficacy of highly active antiretroviral therapy (HAART) [1]. HPgV infections are consistently associated with higher CD4+ cell counts and lower HIV-1 viral loads [5,8,18,32]. This suggests the presence of a significant interaction between HPgV and HIV-1, a relationship that is also supported by the growing evidence for the inhibitory effect of HPgV on HIV-1 replication in vitro [33,34]. These collective findings emphasize the intricate interplay between these two viruses and the potential impact of HPgV on the clinical course of HIV-1 infection. The results of the present study are consistent with previously established findings.

Specially, a significant decrease in HPgV infection rates is observed when HIV-1 RNA viral load increases in patients. This observation suggests an inverse and bi-directional correlation between HIV-1 and HPgV viral load. However, it is essential to acknowledge that this retrospective analysis does not allow for a definitive causal analysis. To gain a comprehensive understanding of the intricate relationship between HPgV and HIV co-infection, further extensive longitudinal studies are required to investigate the profound impact of HPgV on HIV prognosis. Studies on HPgV genetic diversity revealed the existence of seven different genotypes worldwide with significant regional variation [16,29]. The results showed that the genotypic diversity and complexity of HPgV circulating in Istanbul, Türkiye, were not very high. In this study, genotype 2 was identified in almost all samples. The findings of the present study differed from those of the Asian study but were similar to the European and American study results.

This is expected, as in the majority of the countries, just one genotype is prevailing in more than 90% of positive individuals. This is because the HPgV genotype distribution is closely related to the genetic background of the population [35]. Genotype 1 is generally the dominant genotype in the African continent. The fact that the patient who was observed to have genotype 1 was an individual from Africa coincides with this information. The relationship between HIV and HPgV co-infection needs to be addressed in future long-term studies to investigate the effect of HPgV on HIV prognosis.

## 5. Conclusions

The findings reveal a substantial prevalence of HPgV infection (27.3%) among HIV-infected patients in Istanbul, which is notably higher than observed in some other regions. The predominant genotype distribution of HPgV in this population was found to be genotype 2, particularly subtypes 2a and 2b. These results are consistent with previous research demonstrating the potential beneficial effects of HPgV on HIV disease progression, characterized by higher CD4+ cell counts and lower HIV-1 viral loads in co-infected individuals. The study also highlights an inverse correlation between HIV-1 and HPgV viral loads, suggesting a potential interaction between the two viruses. However, it is important to emphasize that this study provides observational data, and further research, including extensive long-term studies, is required to clarify the relationship between HPgV and HIV co-infection. These future investigations will be crucial to comprehensively understanding the impact of HPgV on the prognosis and management of HIV-infected individuals. Overall, this study contributes valuable insights into the prevalence and genotypic diversity of HPgV in HIV-infected patients in Türkiye and highlights the need for ongoing research to explore the clinical implications and potential therapeutic opportunities associated with HPgV in the context of HIV co-infection.

## Figures and Tables

**Figure 1 viruses-16-00005-f001:**
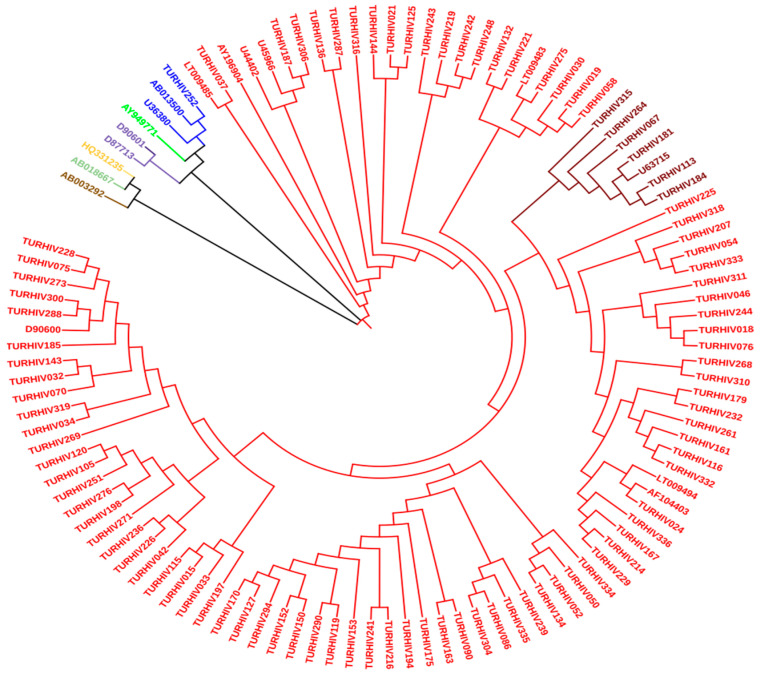
Phylogenetic analysis of 96 nucleotide sequences of HPgV among HIV-1-infected patients from Istanbul, Türkiye. AB013500, U36380 (Genotype 1); D90600, AY196904, U44402, U45966, LT009485, LT009494, AF104403 (Genotype 2; 2a); U63715 (Genotype 2; 2b); D87713, D90601 (Genotype 3); AB0188667 (Genotype 4); AY949771 (Genotype 5); AB003292 (Genotype 6); and HQ331235 (Genotype 7).

**Figure 2 viruses-16-00005-f002:**
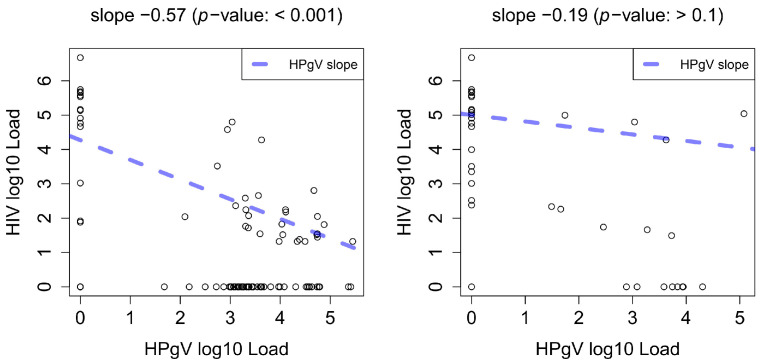
Scatter plots of HIV-1 and HPgV load. The slope (blue) was fitted by regressing HIV-1 viral load on HPgV viral load and other covariables such as age, CD4 and CD8 counts, and CD4/CD8 ratio. The slope shows a linear relationship between both viruses. Left is the regression of the cross-sectional data and right is the regression of the longitudinal data. The longitudinal linear relationship had a lower effect size, and the relationship between HIV-1 and HPgV load was more dichotomous without many continuous HIV-1 and HPgV viral load pairs.

**Figure 3 viruses-16-00005-f003:**
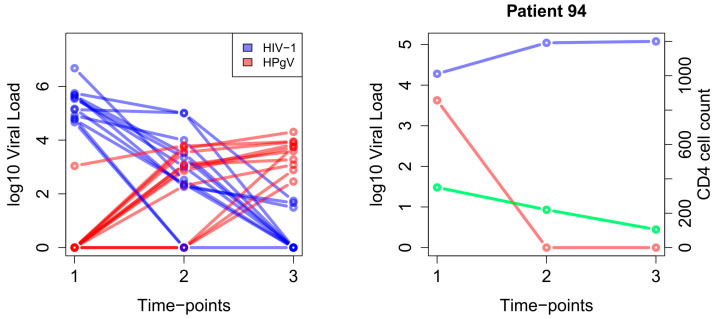
Longitudinal viral loads of HIV-1 (blue) and HPgV (red). The data shows trends of decreasing HIV-1 and increasing HPgV load. On the right is the only patient (patient 94) with an opposing trend of HIV-1 and HPgV loads over time. The CD4 cell count (green) agrees with this trend.

**Table 1 viruses-16-00005-t001:** Comparison of variables in HIV-infected patients with and without HPgV.

Demographics	HPgV-Negative *n* = 255 (72.6%)	HPgV-Positive *n* = 96 (27.3%)	*p* Value
Sex			0.12
Male	221 (86.7)	91 (94.8)	
Female	34 (13.3)	5 (5.2)	
Age			1
18–30 years	40 (15.7)	16 (16.7)	
31–50 years	145 (56.9)	54 (56.2)	
≥51	70 (27.4)	26 (27.1)	
Mean age	43.5	41.9	
Mean CD4+ cells/mm^3^	399	486	0.0002 *
<200	74 (29)	8 (8.3)	
≥200	181 (71)	88 (91.7)	
Log of HIV-1 RNA viral load/mL	5.28	4.89	
ART use			0.22
Yes	197 (77.3)	83 (86.5)	
No	58 (22.7)	13 (13.5)	
Mean CD8 + cells/mm^3^	622	743	

* Statistically significant (*p* < 0.05).

## Data Availability

Data are available within the article.

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
