# Peer review of "Interplay Between HIV and Human Pegivirus (HPgV) Load in Co-Infected Patients: Insights from Prevalence and Genotype Analysis"

_viruses, 2023, doi:10.3390/v16010005_

Round 1

Reviewer 1 Report

Comments and Suggestions for Authors

The manuscript Interplay between HIV and human pegivirus (HPgV) load in co-infected patients: Insights from prevalence and genotype analysis shares some interesting data. The introduction gives enough information for the reader and the methods are adequately described. However, there are some issues in the results section; for example, in Table 1, when calculating the percentage of negative vs. positive between genders, the authors should, in my opinion take into consideration that there are far more men than women to begin with, and present results accordingly: roughly 13% women are positive to pegivirus, and about 30% of men. Same goes for the age groups, CD4+ count, therapy... Plots per patient are not very informative, and I think the authors should focus on the one patient that has a different trend (patient 11) and show the rest on the same plot. To conclude, the result presentation needs some corrections, after that the manuscript will be suitable for publishing.

Comments on the Quality of English Language

English language requires moderate editing, perhaps an English language professor or a native speaker should check the manuscript. 

Author Response

Comment: There are some issues in the results section; for example, in Table 1, when calculating the percentage of negative vs. positive between genders, the authors should, in my opinion take into consideration that there are far more men than women to begin with, and present results accordingly: roughly 13% women are positive to pegivirus, and about 30% of men. Same goes for the age groups, CD4+ count, therapy... Plots per patient are not very informative, and I think the authors should focus on the one patient that has a different trend (patient 11) and show the rest on the same plot.

Reply: We added the values describing the disproportion between groups to the main text on page 6. We now placed Figure 4 as new supplementary Figure 1 and amended Figure 3 according to the reviewer’s suggestions. We also renamed patient 11 to patient 94 (as correctly listed in the supplementary Excel list).  

Reviewer 2 Report

Comments and Suggestions for Authors

This is a well written analysis of HPgV infection among HIV+ individuals in Istanbul. It would have been vastly improved if a comparison group of HIV- individuals tested for HPvG using the same methodology were included. There is some reference to earlier studies showing the prevalence of HPgV in Turkiye, but those appear to be very specific populations, located in different geographic areas, and the methodology may not be the same. Thus those prevalence estimates may not be comparable to the ones produced here.

The relatively high prevalence of ongoing infection with HPgV virus in the HIV population, and even in the non-HIV+ Turkish populations raises the question of persistent vs recurrent infection. This is addressed by the inclusion of some longitudinal data, but the authors may wish to discuss this more.

Has a multiple test correction (eg Bonferroni) been applied to the calculation of p values (especially in Table 1)? If so it should be stated in the statistics section, if not it should be applied, p values recalculated, and the test used noted in the statistics section,

Comments on the Quality of English Language

Fine

Author Response

Comment: Has a multiple test correction (eg Bonferroni) been applied to the calculation of p values (especially in Table 1)? If so it should be stated in the statistics section, if not it should be applied, p values recalculated, and the test used noted in the statistics section.

Reply: We thank the reviewer for this suggestion and appropriately corrected the p-values in Table 1 with Bonferroni as stated in the statistics section on page 4.